# Intraocular and intracranial pressure in glaucoma patients taking acetazolamide

**Allison R. Loiselle**[1], **Emile de Kleine**[2], **Pim van Dijk**[2], **Nomdo M. Jansonius**[1]*

**1** Department of Ophthalmology, University of Groningen, University Medical Center Groningen, Groningen, The Netherlands, **2** Department of Otorhinolaryngology/Head and Neck Surgery, University of Groningen, University Medical Center Groningen, Groningen, The Netherlands

* n.m.jansonius@umcg.nl

**Data Availability Statement:** All relevant data are within the manuscript and its Supporting Information files.

**Funding:** N.M. Jansonius received the funding for this study from the European Union (EU) Horizon

## Abstract

The alternative mechanical theory of glaucoma, in which an increased pressure difference across the lamina cribrosa (difference between intraocular and intracranial pressure; IOP and ICP), rather than solely an elevated IOP, leads to structural and functional vision loss, is still controversial. If the theory is true, a drug that simultaneously lowers both the IOP and ICP may be ineffective. The aim of this study was to determine how acetazolamide (AAZ; a drug prescribed in glaucoma that aims to lower the IOP) affects both IOP and ICP in glaucoma patients and to compare the magnitude and time course of the induced pressure changes with those of healthy subjects not taking AAZ. IOP and noninvasive ICP (measured through emissions from the ear) were measured in 20 glaucoma patients taking 125 mg of AAZ twice daily. Measurements were taken for 30 minutes before taking the drug and for 2 hours post-ingestion. Comparisons were made with 13 age-similar controls. After 12 hours with no anti-glaucoma medication, AAZ did not further reduce IOP in glaucoma patients compared to controls (P = 0.58) but did reduce ICP compared to controls (P = 0.035), by approximately 4 mmHg. Our findings suggest that there are periods during the day when the pressure difference across the lamina cribrosa is larger in case of AAZ use. Future studies should focus on improving the noninvasive ICP testing, different doses and dosing schedules of AAZ, and the time course of IOP in glaucoma patients not taking AAZ.

## Introduction

Glaucoma is a chronic and progressive eye disease characterized by loss of retinal ganglion cells, thinning of the retinal nerve fiber layer, and subsequent visual field loss. If left untreated, it can eventually lead to blindness. Currently, a high intraocular pressure (IOP) is the only treatable factor in the pathophysiology of glaucoma. However, many patients continue to progress after IOP is controlled and others have normal-tension glaucoma (NTG) in which their IOP is normal even before treatment [1], suggesting another mechanism is needed to explain the disease. One possible theory is that glaucoma patients have a reduction in intracranial pressure (ICP) [2–4]. But this has not been confirmed in some more recent studies [5–7].

2020 grant No 661883 European Glaucoma Research Training Program (EGRET) and by Uitzicht grant No 2015-11 (ANVVB, Glaucoomfonds, LSBS, Novartisfonds, Oogfonds, Stichting Blinden-Penning). The funders had no role in study design, data collection and analysis, decision to publish, or preparation of the manuscript.

**Competing interests:** The authors have declared that no competing interests exist.

The lamina cribrosa is a porous layer at the back of the eye through which nerve fibers run. In the retrobulbar space, cerebrospinal fluid surrounds the optic nerve and ICP is therefore transferred to this area and can act on the lamina cribrosa. The idea behind the ICP theory of glaucoma is that both the IOP and ICP interact at the posterior part of the eye and, if not in balance, can cause mechanical stress and therefore nerve fiber damage [8]. This balance can be investigated by measuring the trans-lamina cribrosa pressure difference (TLCPD), or the difference between the IOP and the ICP at the level of the lamina cribrosa.

Acetazolamide (AAZ) is a carbonic anhydrase inhibitor that is used in glaucoma treatment to lower IOP, by a mechanism of lowering aqueous humor production [9–12]. However, it is also used in the treatment of high ICP, because it inhibits enzymes in the choroid plexus and decreases production of cerebrospinal fluid [13–16]. If it is true that the TLCPD is a causal factor in the incidence or progression of glaucoma, then a drug that simultaneously lowers both IOP and ICP may not be effective. In fact, if the magnitude of the ICP change is *more* than that of the IOP, it could actually be harmful.

While research on the effects of AAZ on IOP exists [17–22], unfortunately, there is little information about the effects of AAZ on ICP. In the majority of the neurological clinical cases, patients' doses are increased until complaints—like headache or double vision—are alleviated. There is an assumed reduction in ICP, but the absolute change and time course are not well elucidated. Although some research has been completed [16,23,24], the effect of AAZ on ICP on the scale of hours needs further investigation. This can be difficult because the current gold standard for ICP measurement is the lumbar puncture, which is cumbersome and painful for the patient and is not optimal for continuous measurement. Noninvasive methods of ICP measurement like distortion product otoacoustic emissions (DPOAEs), which can be measured continuously, are therefore of great interest.

DPOAEs are emitted by the inner ear in response to two tones at specified levels and frequencies. The emissions are dependent on ICP because of the connection between the fluid in the intracranial space and the inner ear fluid through the endolymphatic and cochlear ducts. Changes in ICP cause pressure changes on the stapes and therefore on the transmission of sound through the middle ear. DPOAEs have already been shown to accurately represent ICP [25–28]. Although DPOAEs are not yet able to give absolute pressure readings, they can show changes in ICP, due to changing body positions or the use of drugs like AAZ. We recently calibrated the relationship between DPOAE change and ICP change by comparing the body-position dependency of DPOAEs with that of ICP [5,6].

If AAZ lowers the IOP and ICP such that the TLCPD remains unchanged, it may be an ineffective treatment for glaucoma. The aim of the current study is therefore to determine the TLPCD in glaucoma patients taking AAZ by measuring IOP and ICP noninvasively and comparing the magnitude and time course of the induced pressure changes with those of healthy subjects not taking AAZ.

## Methods

### Study population

Subjects with healthy eyes who responded to our advertisement and glaucoma patients who were selected based on the inclusion criteria (listed below) from the Groningen Longitudinal Glaucoma Study database [29], received an information letter and informed consent form. The ethics board of the University Medical Center Groningen (UMCG) approved the study protocol. All participants provided written informed consent. The study followed the tenets of the Declaration of Helsinki.

The methods of this study were similar to a previous study from our lab [6]. In short, in order to be eligible to participate in this study, all subjects had to meet the following inclusion criteria: 50 to 70 years of age and detectable DPOAEs in at least one ear. Additionally, for controls: upright IOP of 21 mmHg or lower, no eye disease, and no family history of glaucoma (determined by a questionnaire). To exclude eye disease, we performed optical coherence tomography (OCT-HS100; Canon, Tokyo, Japan; considered normal if the mean retinal nerve fiber layer and retinal ganglion cell layer thickness in macular area were above the 5th percentile), frequency doubling technology (FDT; Carl Zeiss, Jena, Germany; no reproducibly abnormal test locations allowed in C20-1 screening mode), and a measurement of visual acuity (visual acuity at least 0.8 in both eyes). For patients, glaucoma was defined according to Heeg et al [29]. We required a reproducible (same hemifield and at least partially overlapping) visual field defect (Humphrey Field Analyzer 30–2 SITA fast; Carl Zeiss, Jena, Germany; criterion: 'glaucoma hemifield test' outside normal limits) in at least one eye that had to be compatible with glaucoma and without any other explanation. In contrast to our previous study, the patients could include those with primary open angle (POAG), secondary, and pigment dispersion glaucoma. All patients took 125 mg of AAZ for the study, and had a minimum of 12 hours since their last regular dose, which was 125 mg twice daily in all cases. Any IOP-lowering eye drops were continued but the morning dose, if applicable, was delayed until the end of the study-related measurements, which were performed around the morning dose of AAZ (see below for details).

## DPOAE parameters

DPOAEs were measured using hardware (Elios) and software (Echosoft version 2.1.8.1) developed by Echodia (St. Beauzire, France). In order to ensure the highest magnitude at the $2f_1$-$f_2$ emission, we used a fixed rate of $f_2/f_1 = 1.20$ with tones at frequencies $f_1 = 1000$ Hz and $f_2 = 1200$ Hz and levels $L_1 = L_2 = 70$ dB SPL. All DPOAE measurements were completed in a sound-isolated audiometric room in the otorhinolaryngology clinic.

## Measurement protocol

Blood pressure (Omron Model M6 Comfort, Omron Healthcare Co., Ltd.) and upright IOP in both eyes (iCare Pro tonometer, Icare Finland Oy) were measured. Subjects then laid down on an airbed and IOP was measured once again in the supine position.

For DPOAEs, the ear with the highest signal to noise ratio (SNR) was used for testing. DPOAEs were measured after 1 minute in the supine position, and then subsequently every 15 minutes. After 30 minutes in the supine position, patients were asked to take their AAZ, and healthy subjects were asked to simply sit up and take a sip of water. The three measurements taken before drug administration were averaged to create a baseline. After this, measurements resumed in supine position every 15 minutes for the remaining 2 hours. For each test, 10 DPOAE measurements were taken over approximately 30 seconds. IOP was measured immediately after each DPOAE test. Blood pressure was also measured 30 and 90 minutes after AAZ intake for patients and after the healthy subjects sat up to drink water.

## Data analysis

The subject characteristics were described with mean and standard deviation (SD) for normally distributed variables. For variables with a skewed distribution, we used median and interquartile range (IQR) instead. Groups were compared using the student's t-test and proportions were compared using a chi-square test.

The stimulus and data collection protocols have already been described in detail [30]. From the 10 DPOAE measurements taken at each 15 minute interval, those with a SNR of less than 3 dB were excluded after which the remaining measurements were averaged. For healthy subjects and patients with bilateral glaucoma, the IOP of both eyes was averaged. For those with unilateral glaucoma (4 out of 21 patients), only the affected eye was used. All participants started the protocol in the morning between 8:30 and 10:00 to make the inevitable influences of diurnal fluctuations as similar as possible within and between the groups.

Due to the subject specific, unknown offset of the DPOAE phase (the quantity provided by the device), data were presented relative to the calculated baseline value. Next, the resulting DPOAE phase shift was converted into an ICP change from baseline using a conversion factor of 4 degrees/mmHg [6]. From the time series of ICP measurements (each 15 minutes during two hours, with the AAZ intake in patients and the sip of water in controls at t = 0), we determined a trend by using a linear regression for each subject. A one-sided t-test was used to compare the resulting slopes between patients and controls ($H_0$: AAZ has no influence on ICP; $H_1$: AAZ lowers ICP). The difference in the slope of the ICP change between patients and controls was considered to be the AAZ effect on ICP. A similar comparison between the groups was made for IOP. Finally, the ICP slopes were compared to the IOP slopes in the patients, using a two-sided paired t-test ($H_0$: effect of AAZ on ICP equals effect on IOP; $H_1$: effect of AAZ is different for ICP and IOP).

All analyses were performed using R (version 3.3.3; R Foundation for Statistical Computing, Vienna, Austria). A P-value of 0.05 or less was considered statistically significant.

## Results

The characteristics of the study population can be found in Table 1. From the 21 patients and 14 controls tested, one patient and one control had to be removed from the analysis due to low signal to noise ratios during multiple measurements. Therefore, 20 patients and 13 healthy subjects were used for analysis. The groups were age- and gender-similar and, despite the patients stopping all anti-glaucoma drugs for a minimum of 12 hours before the study, there was no difference in upright IOP measurements.

Fig 1 shows the effect of AAZ on DPOAE phase in glaucoma patients compared to controls from baseline until 2 hours post-drug, when the expected peak effect should occur. A negative phase shift means a lower ICP. There was a significant difference in ICP (the AAZ effect on ICP) between the groups (P = 0.035). Given the conversion factor mentioned above, the observed AAZ effect on ICP from baseline to the final 2 measurements of the 2 hour measurement period post AAZ ingestion would equate to approximately a 4 mmHg decrease in ICP for patients compared to controls [6].

**Table 1. Characteristics of the study population.**

|  | Glaucoma n = 20 | Healthy n = 13 | P-value |
|---|---|---|---|
| Age (yrs) | 61.4 ± 5.2 | 59.5 ± 6.6 | 0.38 |
| Gender (% female) | 35.0% | 61.5% | 0.44 |
| Supine SBP (mmHg) | 130.4 ± 13.3 | 122.7 ± 15.7 | 0.30 |
| Supine DBP (mmHg) | 82.6 ± 7.4 | 75.4 ± 10.7 | 0.17 |
| VF MD better eye (dB; median [IQR]) | -13.1 (-4.4 to -18.8) | NA | - |
| VF MD worse eye (dB; median [IQR]) | -25.7 (-14.3 to -27.7) | NA | - |
| Pre-treatment IOP (mmHg; median [IQR]) | 32.0 (25 to 38) | NA | - |
| Upright IOP (mmHg) | 16.6 ± 5.3 | 16.1 ± 3.4 | 0.59 |

*SBP* Systolic blood pressure, *DBP* Diastolic blood pressure, *VF MD* standard automated perimetry mean deviation, *IOP* intraocular pressure.

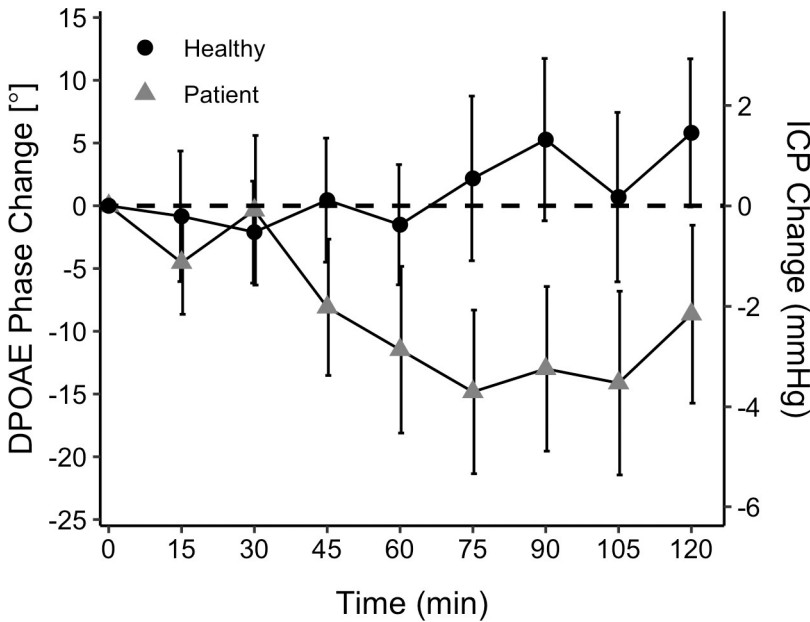

**Fig 1. Effect of AAZ on DPOAE phase (mean ± standard error) in glaucoma patients (n = 20) compared to controls not taking AAZ (n = 13).** ICP on the secondary y-axis was calculated from the DPOAEs based on a previously determined relationship of 4 degrees/mmHg [6]. Time t = 0 was the calculated baseline value and the drug administration time for patients.

Fig 2 shows the effect of AAZ on IOP in glaucoma patients compared to the IOP in controls, not taking AAZ. There was no difference in IOP change between the groups (P = 0.58). Despite the apparent trend that AAZ had a greater effect on ICP than IOP, the AAZ induced

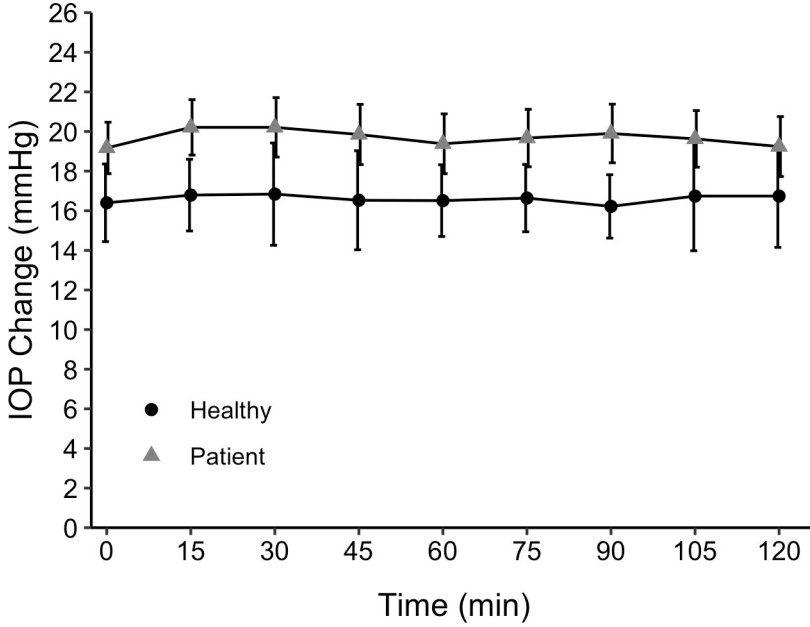

**Fig 2. Effect of AAZ on IOP (mean ± standard error) in glaucoma patients (n = 20) compared to controls not taking AAZ (n = 13).** Time t = 0 was the calculated baseline value and the drug administration time for patients.

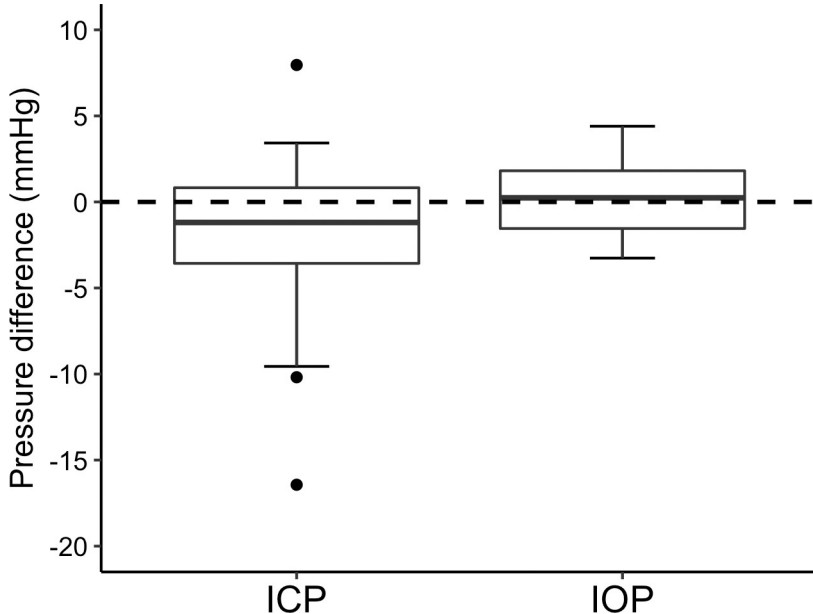

**Fig 3. Difference in ICP and IOP from baseline to 2 hours post-AAZ ingestion in glaucoma patients (n = 20).**

change of ICP versus IOP in patients did not differ significantly (P = 0.085). Fig 3 shows ICP and IOP in patients at 2 hours post ingestion, relative to baseline. The comparison utilizes the previously mentioned 4 degrees/mmHg conversion so that both pressures are expressed in the same unit.

## Discussion

In the current study, we find that after 12 hours with no anti-glaucoma medication, 125 mg of AAZ does not further reduce IOP in glaucoma patients, but does reduce ICP. Controls did not show a reduction, suggesting that the ICP reduction in glaucoma patients indeed is an AAZ effect and not a diurnal one.

The IOP lowering effect of AAZ was already demonstrated 65 years ago when Becker et al [11] showed that IOP was reduced 43.6% in POAG patients within 3–5 hours of taking 250 mg of AAZ. Subsequent studies have shown reductions in IOP between 15–34% in patients with POAG and ocular hypertension within hours after varying doses of AAZ [19,20,22,31,32]. At a dose of 125 mg—the dose used in the current study—IOP was reduced by 29% in OHT patients within 2 hours [19]. In the current study, IOP did not significantly change from baseline to 2 hours after taking AAZ. A major difference between the earlier studies and our study is that in the above-cited studies they used a one-time administration whereas we studied IOP before and after a regular intake. Dose intervals are normally chosen to minimize plasma level fluctuations (with an interval shorter than the half-life of the drug), but this is not the case for AAZ twice daily (half-life of AAZ: 8 hours). If the half-life of a drug equals the dose interval, the plasma level during continuous dosing ranges from $c_{max}$ to $2c_{max}$, with a mean plasma level of $1.44c_{max}$ (assuming first order kinetics), where $c_{max}$ is the peak plasma level (concentration) after a one-time administration. As such, the expected AAZ plasma level fluctuation during continuous dosing of 125 mg every 8 hours would equal the peak plasma level after a one-time administration of 125 mg. Obviously, the effect of a drug is not directly proportional to the plasma level—at some level saturation occurs. A better formulation is therefore that the

expected effect of AAZ on IOP during continuous dosing of 125 mg every 8 hours would equal the difference in effect between a one-time administration of 250 mg and a one-time administration of 125 mg. Given that this difference in effect has been observed [18–20], we had expected to observe an effect of AAZ on IOP in our study design, especially because the plasma level fluctuation of 125 mg every 12 hours should be even larger than that of 125 mg every 8 hours.

Another possible explanation for the limited effect on IOP is that there may have been some influence from the topical IOP lowering drugs that some patients were taking. Patients continued using these drugs uninterrupted (see below), except for delaying any morning dose, if applicable, until the end of the current study. The effect of two IOP-lowering drugs together is smaller than the sum of the individual effects [33] and, for topical medication, this might influence IOP more than ICP. As all but one of our patients used topical IOP-lowering drugs alongside AAZ, we could not test this hypothesis. Finally, glaucoma has been reported to increase diurnal fluctuations in IOP. The IOP pattern during the day varies between and even within individual patients, but, on average, IOP tends to peak in the morning [34,35]. As such, our flat IOP curve in the glaucoma patients might actually reflect a successful treatment of the morning peak. A control group of glaucoma patients not taking AAZ would have been helpful here.

While there is ample literature describing the effects of AAZ on IOP, this is not the case for its effect on ICP. Since lumbar punctures are invasive and painful, there is a lack of data on healthy subjects. Lumbar punctures also do not work well for continuous ICP measurement, so very little is known about hourly changes—both from normal diurnal fluctuations and those induced by medications. The only dose of AAZ at which there was a crossover in available data for changes in IOP and ICP was 500 mg. In a study of 36 patients with cerebrospinal fluid leaks who were given 500 mg of AAZ, there was an ICP reduction of 7.4 mmHg within 4–6 hours of administration [24]. This is a reduction of 31.5%, which is similar to that seen in IOP. In our study, the greatest reduction in DPOAE phase was 14.8 degrees in patients taking AAZ. By using the previously defined ratio of 4 degrees of phase shift per mmHg of ICP change, this would equate to an ICP reduction of 3.7 mmHg. This value makes physiological sense in comparison to the study from Chaaban et al. because the patients in the current study did not start with a high ICP, were given a lower AAZ dose, and got their dose as part of an ongoing treatment with AAZ twice daily.

A limitation in this study was the above mentioned inability to implement a "wash-out" period for patients to make sure that no previously taken IOP lowering medications could affect the results. Stopping their drugs for a longer period was not possible due to the ethical limitations on asking patients to stop treatment. However, measuring IOP and ICP during chronic use of IOP-lowering drugs is more representative of the real life pressure changes experienced by patients. All subjects in this study were taking 125 mg of AAZ twice daily, as this is the most common dose given at our clinic for a long period of time. It is possible that higher doses may have elicited a change in IOP with or without a more marked reduction in ICP. A strong point in this study is that we were able to use a noninvasive measurement technique to measure IOP and ICP in glaucoma patients for the first time, at an unprecedented resolution in time. However, while the test was simple and comfortable for patients, the intersubject variability in DPOAE phase responses may limit the ability to accurately assess ICP changes in individual patients.

What are the clinical implications of our observations? If ICP does indeed play a role in the pathophysiology of glaucoma, which would make the TLCPD the target for treatment, then the ICP lowering effects of AAZ may be counterproductive, when IOP remains constant. In this study, we showed that ICP may fluctuate by about 4 mmHg even at an AAZ dose of only

125 mg twice daily. This would suggest that while IOP reaches a plateau during treatment with AAZ, ICP may dip hours after ingestion and create periods during the day when the TLCPD is larger. However, it could also be the case that the TLCPD with chronic AAZ therapy is lower than without, except for a few hours after the AAZ dose. The mean effect of AAZ on IOP would then exceed the mean effect on ICP but with less fluctuation for IOP.

In conclusion, this study showed that glaucoma patients taking AAZ long-term did not have a further IOP reduction immediately after taking their normal AAZ dose of 125 mg, however, their ICP did decrease during this time compared to healthy subjects. Future studies should focus on improving the efficacy of noninvasive ICP testing and on absolute measurements. Results both with and without a wash-out period are needed, results with other AAZ doses and dosing schedules, and results in glaucoma patients not taking AAZ, to better determine if AAZ is helpful or harmful in glaucoma.

## Supporting information

**S1 Data.**
(XLSX)

## Acknowledgments

We would like to thank F. Aanstoot for her assistance with data collection.

## Author Contributions

**Conceptualization:** Allison R. Loiselle, Emile de Kleine, Pim van Dijk, Nomdo M. Jansonius.

**Data curation:** Allison R. Loiselle.

**Formal analysis:** Allison R. Loiselle, Nomdo M. Jansonius.

**Funding acquisition:** Nomdo M. Jansonius.

**Investigation:** Allison R. Loiselle, Emile de Kleine, Pim van Dijk, Nomdo M. Jansonius.

**Methodology:** Allison R. Loiselle, Emile de Kleine, Pim van Dijk, Nomdo M. Jansonius.

**Project administration:** Allison R. Loiselle, Emile de Kleine, Pim van Dijk, Nomdo M. Jansonius.

**Supervision:** Emile de Kleine, Pim van Dijk, Nomdo M. Jansonius.

**Validation:** Allison R. Loiselle.

**Visualization:** Allison R. Loiselle, Nomdo M. Jansonius.

**Writing – original draft:** Allison R. Loiselle, Nomdo M. Jansonius.

**Writing – review & editing:** Allison R. Loiselle, Emile de Kleine, Pim van Dijk, Nomdo M. Jansonius.

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
