## [Decision Letter · Decision Letter 0]

2 Jun 2020

Intraocular and intracranial pressure in glaucoma patients taking acetazolamide

PONE-D-20-14239

Dear Dr.  Loiselle,

We are pleased to inform you that your manuscript has been judged scientifically suitable for publication and will be formally accepted for publication once it complies with all outstanding technical requirements.

With kind regards,

Sanjoy Bhattacharya

Academic Editor

PLOS ONE

Reviewers' comments:

Reviewer's Responses to Questions

**Comments to the Author**

1. Is the manuscript technically sound, and do the data support the conclusions?

Reviewer #1: Yes

Reviewer #2: Yes

2. Has the statistical analysis been performed appropriately and rigorously? 

Reviewer #1: Yes

Reviewer #2: Yes

3. Have the authors made all data underlying the findings in their manuscript fully available?

Reviewer #1: Yes

Reviewer #2: Yes

4. Is the manuscript presented in an intelligible fashion and written in standard English?

Reviewer #1: Yes

Reviewer #2: Yes

5. Review Comments to the Author

Reviewer #1: The manuscript is well organized, presented in intelligible fashion and written in standard English.

I understand this may not be so relevant to the objective of this study, however in Lines 139,150: the Blood Pressure measurement was done while the participants in Supine position with this device (Omron Model M6 Comfort,) and the instruction manual of that device states that the patient has to be in Chair with Back and arm being supported. So this can result in less accurate readings and in turn, less accurate understanding of the relationship between the AZZ and BP for example.

Also as mentioned it will be very helpful if we can have Glaucoma patients who are not on AZZ.

Reviewer #2: The authors applied a new noninvasive method to measure ICP longitudinally after acetazolamide (AAZ) treatment in glaucoma patients. By comparing the differential effect of AAZ on ICP and IOP reduction, the authors concluded that AAZ treatment may actually increase the trans-lamina cribrosa pressure difference at periods immediately after the treatment.

The manuscript is well-written and the analysis is rigourly conducted, despite acknowleged design limitations which are discussed in great detail by the authors. Although the study does not have a control patient group not taking AAZ, the longitudinal design including pre- and post-treatment measurements circumvents such drawback and still makes the analysis sound.

6. PLOS authors have the option to publish the peer review history of their article (what does this mean?). If published, this will include your full peer review and any attached files.

Reviewer #1: Yes: Nayef K Alshammari

Reviewer #2: No

---

## [Editor Report · Acceptance letter]

9 Jun 2020

PONE-D-20-14239 

Intraocular and intracranial pressure in glaucoma patients taking acetazolamide 

Dear Dr. Loiselle:

I'm pleased to inform you that your manuscript has been deemed suitable for publication in PLOS ONE. Congratulations! Your manuscript is now with our production department. 

Kind regards, 

on behalf of

Dr. Sanjoy Bhattacharya 

Academic Editor

PLOS ONE